# To Compete or To Collude: Builder Incentives in MEV-Boost Auctions

Fei Wu [1]   Thomas Thiery [2]   Stefanos Leonardos [1]   Carmine Ventre [1]

## Abstract

This paper employs empirical game-theoretic analysis (EGTA) to examine builders' incentives for strategic bidding in MEV-Boost auctions under the current Proposer-Builder Separation (PBS) framework. Our results suggest that under the ideal conditions of a builder market that lead to decentralization, builders are incentivized to collude rather than compete, contributing to low efficiency in the MEV-Boost auction. We show that latency advantage incentivizes builders to bid strategically to maximize their profit. Additionally, we show that advantages in private orderflow access can incentivize builders to refuse collusion and dominate the market. Furthermore, we demonstrate that the relay enforcement of rejecting new bids after the beginning of the slot, as a mitigation for timing games, impacts builders' strategic bidding incentives. Through our analyses, we highlight the challenge of creating a decentralized yet competitive builder market.

## 1. Introduction

With the introduction of *Proposer-Builder Separation* (PBS) mechanism in Proof-of-Stake Ethereum, block *proposers* can opt into *MEV-Boost* to outsource the task of block production for *Maximal Extractable Value* extraction to *builders* through trusted intermediaries known as *relays* at an out-of-protocol marketplace, where builders compete in the *MEV-Boost auctions* for the right of block production. The auction for the block of slot $n$ typically starts around the beginning of slot $n-1$ and terminates at the end of slot $n-1$ when the proposer calls getHeader to select the winning bid. The highest bidder wins the auction and, upon their block being selected and proposed, stands to potentially realize profits.

Builders' bid value depends on the block value they produce, which originates from priority fees (gas) and *Maximal*

*Extractable Value* (MEV) of public transactions broadcast in the public mempool and private orderflow secured from searchers. MEV refers to the value that can be extracted by including, excluding, and re-ordering transactions (Daian et al., 2019). To gain a competitive advantage in MEV-Boost auctions, builders optimize their connection to searchers and relays, targeting lower *latency* and higher orderflow access. Some builders "vertically integrate", by operating their own searchers and initiating a relay to further improve latency and gain exclusive access to private orderflow. Such entities are referred to as *integrated builders*, and this strategy is termed *vertical integration*. Vertical integration potentially contributes to centralization in the block construction market; today although more than 30 builders are active in the market, the market has been dominated by only several builders. However, there is limited practical insight into how advantages in latency and orderflow access affect builders' strategic bidding behaviors in MEV-Boost auctions.

Another concern under the current PBS framework is *timing games* (Schwarz-Schilling et al., 2023), which involves strategically delaying the block proposal by proposers and optimizing their profits. Such practices can undermine the blockchain's consensus (Schwarz-Schilling & Nueder). A viable countermeasure could involve relays enforcing stricter timeliness on proposers, such as rejecting builder bids after the beginning of the slot. However, there is notably scarce literature on how these timing game mitigations will affect builders' strategic bidding in MEV-Boost auctions.

In this paper, we address these questions and thereby contribute to bridging the existing knowledge gap concerning the interaction between builders' strategic bidding incentives and the MEV-Boost auction mechanism under the current PBS framework. We use the MEV-Boost auction simulation framework and the bidding strategies proposed by (Wu et al., 2024) and conduct an empirical game-theoretic analysis (EGTA). We study builders' strategic bidding in MEV-Boost auctions under varying scenarios and examine how vertical integration impacts builders' incentives. We further investigate how the relay enforcement policy of rejecting bids after the beginning of the slot affects builders' incentives for choosing strategies. This paper makes the following contributions:

    1. We find that, Under the current PBS framework and

---

[1]Department of Informatics, King's College London, London, United Kingdom [2]Robust Incentives Group, Ethereum Foundation. Correspondence to: Fei Wu <fei.wu@kcl.ac.uk>.

ideal conditions of a builder market (similar latency and orderflow access), builders are incentivized to marginally increase their bids to outbid each other (i.e., *colluding*) rather than bidding their full valuation. Although this collusion potentially enhances decentralization, with builders having an equal chance of winning the auction, the winning bids do not reflect (i.e., are much lower than) the actual block values. As a result, the MEV-Boost auction mechanism does not efficiently capture MEV, and builders retain a large proportion of MEV for themselves.

2. We next study how advantages in latency and private orderflow access affect builders' incentives of adopting different bidding strategies. Our results indicate that lower latency enables builders to bid strategically and maximize their profit. In contrast to the symmetric (idealized) scenario studied above, we also find that differences in latency between builders can serve as a critical element that increases the efficiency of the current MEV-Boost auction mechanism. We further show that significant advantages in private orderflow access incentivize builders to engage in competition to increase their market shares (and dominate the market) rather than colluding.

3. Finally, we study the relay enforcement policy of rejecting bids after the beginning of the slot and demonstrate that it significantly impacts builders' bidding incentives when builders have different latencies. We show that this enforcement, which essentially boils down to builders knowing the exact auction termination time, contributes to enhancing the auction efficiency by forcing competitive players with a latency advantage to bid their full valuation.

## 2. Auction Game Model and Strategy Space

We employ the MEV-Boost auction model and bidding strategies proposed by (Wu et al., 2024). We consider the game as *one* auction and 10 players in the auction game, since currently, the top 10 builders build $98.65\%$ blocks built via MEV-Boost. We consider a strategy space that contains three strategies, which are the *naive* strategy, the *adaptive* strategy, and the *last-minute* strategy. We will qualify players by their strategy; so, e.g., "naive players" are those playing the naive strategy. The payoff of the player is the profit they make through the auction.

The model is calibrated using the mempool data (Flashbots, b) and on-chain data maintained by Flashbots and Dune for the period of February 23 to April 7 2024 (Flashbots & Dune). However, due to the lack of data on private orderflow, we apply an approach of estimation based on on-chain data. The detailed definitions and calibration method can be found in Appendix A.

## 3. Empirical Games

Explicitly solving the above game is computationally challenging, as players can independently decide their bidding strategy, which results in $3^{10}$ distinct strategy profiles. We follow an empirical game-theoretic approach where we exploit certain symmetries between builders to reduce the size of the games. Specifically, each player is characterized by their latency and their (prior distribution of) private orderflow access probability. Accordingly, we analyze the games with three variants in which either one or both of these attributes are uniform across all players. We further use *heuristic payoff table* (HPT) (Tuyls et al., 2020) to store the payoff information. In this way, we replace the games with *symmetric* games or *role-symmetric games*. The detailed game definition can be found in Appendix B.

To solve the above games, we employ the $\alpha$-*Rank* algorithm (Omidshafiei et al., 2019). The solution (equilibrium) is presented by a ranking of strategy profiles with their stationary probabilities within the unique stationary distribution of the $\alpha$-Rank Markov Chain. To convey the equilibrium clearly, we present the results as the average number of players using each strategy across all the profiles weighted by their stationary probabilities, which captures the expected frequency of each strategy being used by players in the long run. In addition, we pay attention to the "optimal" strategy profile, i.e., the highest-ranked profile with the highest stationary probability.

## 4. Empirical Game-Theoretic Analysis

In this section, we present our experimental results using the three game variants. We study builders' incentives for choosing strategies under varying conditions, investigate the impact of latency improvements and orderflow access advantage on builders' incentives, and analyze the state of the MEV-Boost auction in equilibrium.

### 4.1. Adaptive players dominate in the symmetric game

In the empirical games where all 10 players share identical latency and the same prior distribution of private orderflow access probability, the optimal strategy profile, where all 10 players adopt the adaptive strategy, demonstrates a high evolutionary stability, with a stationary probability of 0.99997. The result shows that, once the system reaches this state, transitioning to other states is exceedingly unlikely due to the evolutionary stability of this profile and the success of the strategies involved.

Specifically, in the auction simulation, when all the players employ the adaptive strategy, they increase their bids incrementally with the marginal value $\delta$ simultaneously, contributing to uniform bid values among all builders at the end of the auction. Consequently, the auction terminates with a

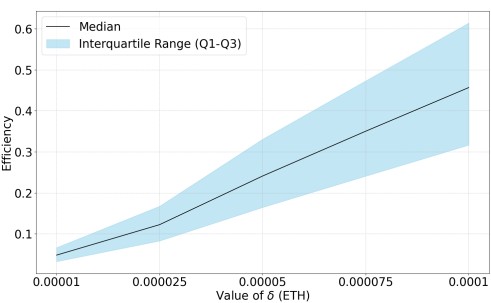

*Figure 1.* Auction efficiencies under varying values of $\delta$ when all 10 builders use adaptive strategy and share an identical latency of 10ms and the prior distribution of private orderflow access.

relatively low value of the winning bid, with each player having an equal opportunity to win the auction. This scenario implies that a significant portion of block value accrues to the winning builder rather than to the proposer, highlighting inefficiencies in the auction mechanism to capture MEV.

To evaluate the auction's capability of capturing MEV, we define *auction efficiency* as the ratio of the winning bid to the total signal value. In the simulation, we use $\delta = 0.0001$ ETH as the marginal value of the adaptive strategy which results in a median auction efficiency of $46.65\%$. It is worth noting that further reduction in the marginal value $\delta$ can lead to even lower winning bid values and efficiency.[1] Figure 1 illustrates the correlation between auction efficiency and the value of $\delta$.

These findings indicate that, in scenarios where all builders experience identical latency and have comparable access to private orderflow, given the defined strategy space, builders are disinclined to bid their full valuation (adopt the naive strategy) to win the auction. Instead, they are incentivized to maximize their profitability by together increasing their bids incrementally with a small margin, sharing an equal chance of winning the auction. This equal opportunity to win might foster an idealized, more decentralized builder market, with builders potentially sharing the market equally. However, this behavior could be perceived as a form of collusion, which, while enhancing decentralization, reduces competitive bidding and contributes to low auction efficiency.

While the builders are incentivized to maximize their profit by colluding, to prevent the proposer from falling back to local production, they need to ensure that their final bid matches or exceeds the maximum extractable value from the public mempool, i.e., public signal—approximately 40% of the total block value. In our simulation, setting $\delta = 0.0001$ ETH results in a bid value corresponding to $46.65\%$ (median) of the total block value, demonstrating that the simulated auctions remain effective under these settings.

---

[1]The varying $\delta$ values have negligible impact on the stationary probabilities of the profile.

## 4.2. Impact of Latency

In the role-symmetric games where all players have the same distribution of private orderflow access but experience two types of latencies, the equilibrium of the previous (symmetric) situation can be disrupted: if all players choose to play the adaptive strategy, the market will not be equally shared anymore, as the high-latency players have a lower chance of winning than the low-latency players. Consequently, high-latency players may be incentivized to switch from the adaptive strategy to the naive or last-minute strategy, as the bidding behaviors defined by these two strategies are not affected by latency.

It is worth mentioning that the adaptive strategy is more sensitive to latency variations, and a higher latency typically results in a slower reaction for the adaptive players. Thus, the primary factor influencing players' strategy choices in this case is the difference in latency between low-latency and high-latency players. To analyze these effects, we consider scenarios under varying latency differences between 5 low- and 5 high-latency players, from 0ms (previous symmetric scenario) to 50ms, and analyze each latency difference scenario as a separate game. Specifically, low-latency players maintain a fixed latency of 10ms, while the latency of high-latency players starts at 10ms and increases by 10ms increments. We then analyze the equilibria of these games to understand the impact of latency on strategy choices.

Our findings show that when the latency difference is only 10ms, the optimal strategy profile remains the one in which all players adopt the adaptive strategy and players are still more incentivized to collude. As suggested by our simulations, despite a lower win rate of 4.45%, high-latency players continue to collude with low-latency players, attracted by the potential to capture approximately 50% of the block value upon winning. However, as the latency difference exceeds 10ms, high-latency players' win rates decline significantly, prompting a strategic shift towards the naive strategy for improved performance.

Figure 2 presents the equilibria computed by $\alpha$-Rank under varying latency differences. As the high-latency players' incentives for switching to the naive strategy become increasingly strong, when the latency difference becomes 20ms, the equilibrium shifts, and the optimal strategy profile changes to all low-latency players adopting the adaptive strategy and all high-latency players adopting the naive strategy. This shift occurs because the effectiveness of the adaptive strategy diminishes significantly for high-latency players as their latency increases, making the naive strategy more appealing. In response to high-latency players' shift to the naive strategy, we also observe a reaction from low-latency players when the latency difference is not greater than 20ms. As the latency difference increases, the optimal profile—where all low-latency players adopt the adaptive strategy and all

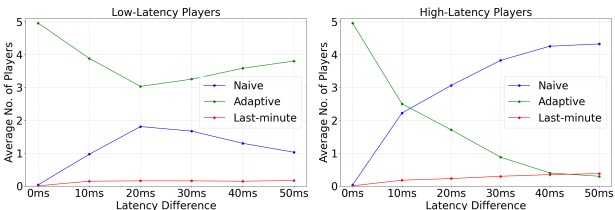
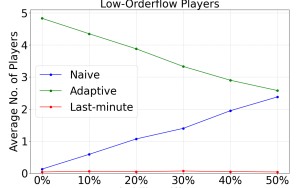
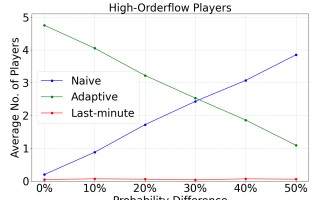

*Figure 2.* Average usage of each strategy by low-latency players (left) and high-latency players (right) across all profiles under varying latency differences as computed by $\alpha$-Rank.

*Figure 3.* Average usage of each strategy by high-orderflow players (left) and low-orderflow players (right) across all profiles under varying probability differences as computed by $\alpha$-Rank.

high-latency players adopt the naive strategy—becomes increasingly dominant. This consistency demonstrates that low-latency players have a strong incentive to maintain the adaptive strategy, while high-latency players are better suited to the naive strategy. Furthermore, when the latency for high-latency players is particularly high, even with a 50% risk of missing the submission window, the last-minute strategy proves more effective than the adaptive strategy. This is attributed to the decreasing effectiveness of the adaptive strategy as latency increases, combined with the occasional effectiveness of the last-minute strategy against the adaptive strategy employed by low-latency players. We also show why the optimal profile is more dominant through a pairwise comparison (Appendix C).

The results underscore the significant impact of latency improvements on builders' incentives for strategic bidding. Although both low-latency and high-latency builders have comparable access to private orderflow in the game settings, latency advantages facilitate faster access to transactions and quicker bid updates. This capability proves crucial near the auction termination because if a late transaction occurs, low-latency builders can include it and update their bids before the auction closes, unlike high-latency builders who, despite having access to the same transaction, cannot update their bids in time. Thus, builders who benefit from a latency advantage are incentivized to adopt the adaptive strategy, thereby maximizing their profits by marginally outbidding. Conversely, builders with a latency disadvantage are compelled to bid their full valuation to enhance performance and offset their latency disadvantage.

Furthermore, the results indicate that the latency difference between builders under the current PBS framework serves as a crucial element that makes MEV-Boost auction efficient. Although high-latency builders may win infrequently, their incentive for naive bidding behavior still pressures low-latency adaptive players to place higher bids, thereby enhancing the auction efficiency.

### 4.3. Impact of Orderflow Access

We next proceed to analyze builders' incentives when their orderflow access probabilities are different (i.e., when these probabilities are drawn from different distributions). To isolate the effects of private orderflow access, we mirror our previous approach in the study of latency, and examine scenarios involving 5 low- and 5 high-orderflow players, all with the same latency.

Surprisingly, despite varying differences in the orderflow access probability between low- and high-orderflow players, the equilibrium outcomes we observe are consistent with the initial game where all players have the same distribution of orderflow access probability: all players choose to play the adaptive strategy. This consistency arises because, under the current simulation settings, the ultimate bid value approximates the public signal value, as discussed in Section 4.1. As a result, both high-orderflow and low-orderflow players remain competitive at the ultimate bid value, rendering the differences in orderflow access inconsequential.

Nevertheless, there exists a scenario where the high-orderflow players are incentivized to adopt the naive strategy and dominate the market without sharing it with the low-orderflow players. This is the case, when builders can dynamically adjust their profit margin based on their private orderflow volume. To study this effect, instead of having a fixed profit margin value symmetric for all players, the profit margin of each player is set to be equal to 50% of their private orderflow volume, i.e., 50% of their private signal. Similarly, we analyze each probability difference scenario as a separate game. We set the orderflow access probability of the low-orderflow players to 50% and increase the high-orderflow players' access probability in 10% increments.

Figure 3 presents the equilibria for various parameter values (orderflow access probability differences) in the above scenario. As we see, high-orderflow players are more incentivized to adopt the adaptive strategy and collude with low-orderflow players when their orderflow access probability and, hence, their profit margin, are low (difference below 30%). However, as their orderflow access probability and, thereby their profit margin, increase, they are increasingly incentivized to adopt the naive strategy. Thus, they increasingly refuse to collude with low-orderflow players and capture a higher win rate. When the probability difference exceeds 30%, meaning high-orderflow players' access

probability surpasses 80%, we observe a significant shift in equilibrium.

For low-orderflow players, their incentive to maintain the adaptive strategy remains strong. Given their limited access to orderflow, adopting the naive strategy is suboptimal because high-orderflow players can easily outbid them. Consequently, their best chance to win is by playing the adaptive strategy and colluding with high-orderflow players. However, as high-orderflow players switch to the naive strategy and refuse to collude, low-orderflow players adopting the adaptive strategy are unable to outbid the high-orderflow players, meaning that the adaptive strategy and the naive strategy are equally effective. Thus, we observe the usage of these two strategies by low-orderflow players tends to converge as the probability difference increases.

## 5. Impact of Relay Enforcement

In this section, we investigate how the relay enforcement policy of rejecting new bids after the beginning of the slot affects builders' incentives for choosing strategies, by fixing the simulated auction interval to match exactly the 12-second duration of a slot (i.e., $\sigma = 0$). We assume that the relay conducts this enforcement policy honestly. This enforcement effectively terminates the auction at the start of the slot, ensuring no further bids are accepted and eliminating the proposer's incentive for delaying their `getHeader` request. It is worth noting that the proposer is not economically incentivized to call `getHeader` earlier, as they might miss a higher bid. Therefore, the builders know that the auction will be terminated at a fixed time point, i.e., the beginning of the slot.

As it turns out, the lack of ambiguity in the termination of the auction significantly affects builders' incentives for choosing strategies. Under non-random termination, the last-minute players, who no longer face a 50% chance of out-of-time revelation, will ultimately bid their valuation at the end of the auction, similar to naive players. This strategy is particularly effective against the adaptive strategy, as it reveals the valuation at the final moment, thereby denying adaptive players any opportunity to react.

We revisit the previous three games under the assumption that the auction interval terminates at exactly 12 seconds. In the first game, where players have equal latency and prior orderflow access distribution, we find that the profile where all players adopt the adaptive strategy continues to dominate with a stationary probability of 0.99506. In the games where players are divided into high-orderflow players and low-orderflow players, the relay enforcement also has a limited impact, as high-orderflow players are able to dominate the market with both naive and last-minute strategies.

However, the equilibria of the games where players expe-

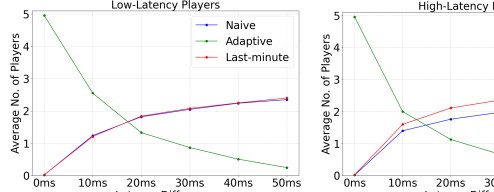

*Figure 4.* Average usage of each strategy by low-latency players (left) and high-latency players across all profiles with the relay enforcement policy under varying latency differences as computed by $\alpha$-Rank.

rience different latencies, in which low-latency players are incentivized to adopt the adaptive strategy and high-latency players are incentivized to adopt the naive strategy, are disrupted. When the auction interval is deterministic, high-latency players using the last-minute strategy can reveal their valuation at the final moment, undermining the effectiveness of the adaptive strategy employed by low-latency players. This forces low-latency players to consider switching to either the naive or last-minute strategies to remain competitive.

Figure 4 displays the experimental results. Similar to the scenarios with varying auction intervals, all players are still more incentivized to collude by adopting the adaptive strategy when the latency difference does not exceed 10ms. However, as players' latency increases, the adaptive strategy becomes less effective for high-latency players, who then favor the last-minute strategy due to its potential to disrupt the adaptive bidding of low-latency opponents. This strategic shift prompts even the low-latency players to adopt either the naive or last-minute strategies, as the effectiveness of the adaptive strategy diminishes in response to increasing latency differences and the heightened incentive for high-latency players to utilize the last-minute strategy.

While the relay enforcement has limited effect on builders' incentives for collusion under ideal conditions as expected, it contributes to offsetting the negative impact on the auction efficiency caused by the latency asymmetries between builders, by forcing low-latency builders to abandon marginally outbidding and bid their full valuation. This is due to the increased effectiveness of the last-minute strategy which allows the high-latency builders to compete and curb inequalities. In turn, this has a noticeable effect on enhancing auction efficiency.

To study this effect, we further compare the auction efficiency between the most robust states of the games under a fixed auction interval and a varied auction interval: 1) low-latency players adopting the naive strategy and high-latency players adopting the last-minute strategy under relay enforcement, i.e., the optimal strategy profile when the auction interval is fixed to 12 seconds; and 2) low-latency players adopting the adaptive strategy and high-latency players

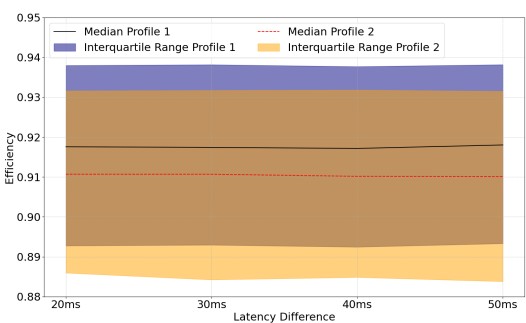

*Figure 5.* Auction efficiency at the most robust state of the games under a fixed auction interval (Profile 1) and a varied auction interval (Profile 2).

adopting the naive strategy under a varied auction interval, i.e, the optimal strategy profile when the auction interval is normally distributed around 12 seconds (as discussed in Subsection 4.2). Figure 5 presents the simulation results, in which Profile 1 refers to the first scenario and Profile 2 refers to the second scenario. We show that, under varying latency differences between the low- and high-latency players, the auction efficiency at the most robust states of the games is enhanced with the relay enforcement.

## 6. Conclusion

Before we conclude the paper, we discuss the limitations and future work in Appendix D.

In this paper, we explore builders' incentives for strategic bidding in MEV-Boost auctions through empirical game-theoretic analysis. Our findings indicated that under ideal conditions of a builder market that leads to decentralization, builders are incentivized to collude by marginally outbidding each other rather than competing by bidding their true valuation, resulting in a low auction efficiency for MEV capture. We demonstrated how advantages in latency and private orderflow access influence builders' strategic bidding. Builders can marginally outbid to maximize their profits with a latency advantage and can refuse collusion to dominate the market with greater orderflow access. Finally, we showed that the relay enforcement policy of rejecting new bids after the beginning of the slot impacts builders' strategic bidding incentives, as it provides the certainty of a fixed auction termination time.

The above results contribute to the ongoing discussion about the challenge of creating a decentralized yet competitive and efficient market (Lu) and provide evidence that current market structures and mechanisms may require fundamental changes to address centralization and efficiency concerns. We highlight the need for further research into block-building auction mechanisms that discourage collusion and encourage genuine competition among builders.

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

# A. Model, Strategies, and Payoff

We employ the MEV-Boost auction model and bidding strategies proposed by (Wu et al., 2024).

We consider a set of $N = \{1, \ldots, n\}$ builders competing in the MEV-Boost auction game. Each builder, indexed by $i$, employs a bidding strategy $s_i$ which can be described as a function $\beta_{s_i} : X \to \mathbb{R}_+$ so that the bid of player $i$ at time $t$ is $\beta_{s_i}(x_{i,t})$, where $x_{i,t} \in X$ represents a vector of input variables at time $t \geq 0$. Whenever redundant, we will omit the dependence on $s_i$ and $x_{i,t}$ and simply write $\beta_{i,t}$. These inputs are discussed next, where we use the terms *player* and *builder* interchangeably.

- **Public signal** $P(t)$**.** The public signal represents the maximum extractable value from public transactions broadcast in the mempool at time $t$, accessible to all builders. New pending transactions are submitted to the mempool as the auction advances. This process is modeled by a compound Poisson process, where the number of transactions $N(t)$ up to time $t$ follows a Poisson distribution with rate $\lambda_p$ and each transaction's value, $V_j$, is randomly drawn from a log-normal distribution. The public signal, $P(t)$, is the cumulative sum of values of $N(t)$ transactions, given by the equation:

$$P(t) := \sum_{j=1}^{N(t)} V_j, \text{ where } N(t) \sim \text{Poisson}(\lambda_p \cdot t) \text{ and } V_j \sim \text{Log-normal}(\xi_1, \omega_1).$$

- **Private signal** $E_i(t)$**.** The private signal represents the *private orderflow* secured from searchers. However, builders often receive similar orderflow because some transactions are commonly shared among them through OFAs, and searchers typically send their bundles to multiple builders. To account for the exclusiveness and correlation of orderflow among players, we introduce an *orderflow access probability*, $\pi_i \in [0, 1]$, for each player $i \in N$, to represent that player's probability of accessing each transaction. The probabilities, $(\pi_i)_{i \in N}$, remain constant throughout the auction interval. Similar to the public signal, the number of private transactions $N_i(t)$ accessed by player $i$ up to time $t$ follows a Poisson distribution with rate $\lambda_e$ but is also influenced by $\pi_i$. Each transaction's value $O_j$ is randomly drawn from a log-normal distribution. The private signal, $E_i(t)$, of player $i$ is

$$E_i(t) := \sum_{j=1}^{N_i(t)} O_j, \text{ where } N_i(t) \sim \text{Poisson}(\lambda_e \cdot t \cdot \pi_i) \text{ and } O_j \sim \text{Log-normal}(\xi_2, \omega_2).$$

$E(t)$ denotes the total value of private orderflow, during the slot at time $t$. Thus, the *aggregated signal*, $L_i(t)$, of player $i$ and the *total signal*, $L(t)$, at time $t$ can be given by combining the public signal and the private signal:

$$L_i(t) := P(t) + E_i(t) \text{ and } L(t) := P(t) + E(t).$$

Given the positive correlations between bid arrival times and bid values (Wahrstätter et al., 2023), we assume that all MEV opportunities are persistent throughout the auction. We also assumed that the MEV of private orderflow secured from searchers is uniform between those who share the private transactions.

- **Latency:** $\Delta_i$**.** The *latency*, $\Delta_i > 0$, of each player depends mostly on that player's network connectivity and geographic location. It quantifies the delay in the relay's acceptance of bids relative to the player's access to a signal update and their subsequent bid submission. It is assumed to be known and constant during the auction and to only affect the player's bidding action.

- **Profit margin:** $pm_i$**.** The profit margin, $pm_i > 0$, quantifies player $i$'s risk tolerance and profit expectations.

- **Valuation:** $v_i(t)$**.** The *valuation* of player $i$ represents the highest bid player $i$ can place at time $t$ while ensuring a positive profit, and is defined as: $v_i(t) := L_i(t) - pm_i$.

- **Current highest bid:** $\max_{j \in N} \{\beta_{j,k}\}_{k \leq t}$**.** This variable represents the highest bid among all bids submitted by all builders up to time $t$. This information is known to all players.

- **Auction termination time:** $T$**.** The auction interval is defined as $[0, T]$, where $T$ denotes the time when the proposer calls getHeader and selects the highest winning bid. Instead of $T$ being exactly equal to 12 seconds as expected, the winning bid is typically selected around $T = 12$ seconds due to factors such as latency or timing games. Thus, $T$ is randomly drawn from a normal distribution with mean 12 and standard deviation $\sigma$.

## A.1. Strategies and payoffs

We consider a strategy space $S$ that contains three strategies, $S = \{s_0, s_1, s_2\}$, where $s_0$ represents the *naive* strategy, $s_1$ represents the *adaptive* strategy, and $s_2$ represents the *last-minute* strategy. We will qualify players by their strategy; so, e.g., "naive players" are those playing the naive strategy. These strategies are defined as follows.

Naive players consistently bid their valuation. Adaptive players either incrementally exceed the highest existing bid by a marginal value $\delta > 0$, or bid their valuation when surpassing the current highest bid is unfeasible. Last-minute players hold their bids initially and bid their valuation after the *revealing time* $0 < \theta \leq T$. To distinguish last-minute players who reveal their valuation very early from naive players, we let the valuation of last-minute players be revealed from the expected auction termination, i.e., $\theta = 12 - \Delta_i$. The bidding behaviours of these strategies are summarised in Table 1, where we use $v_i(t)_+$ to denote the positive part of $v_i(t)$, i.e., $v_i(t)_+ := \max\{v_i(t), 0\}$.

*Table 1.* Bidding strategies.

| Strategy | Bid value at time $t \leq T$ |
|---|---|
| Naive | $\beta_{s_0}(x_{i,t}) = v_i(t)_+$ |
| Adaptive | $\beta_{s_1}(x_{i,t}) = \min\{v_i(t), \max_{j \in N}\{\beta_{j,k} : k \leq t\} + \delta\}_+$ |
| Last-minute | $\beta_{s_2}(x_{i,t}) = v_i(t)_+ \times \mathbf{1}\{t \geq \theta\}$, where $\theta = 12 - \Delta_i$. |

*Remark* A.1. Together with the assumption that $pm_i$ is non-negative for all builders $i \in N$, the above definitions implicitly assume that builders are not willing to win the auction at a negative profit. However, in current practice, builders are willing to subsidize their blocks and win the auction at a net loss (Yang et al., 2024). The main reason that we introduce this assumption is that we consider static (non-repeated) auction games.

We consider 10 players in the auction, since currently, the top 10 builders build $98.65\%$ of the total blocks built via MEV-Boost. Each player $i \in N = \{1, \ldots, 10\}$, selects a pure strategy $s_i \in S$, and bids according to their chosen strategy throughout the auction interval. The collection of strategies selected by all players forms a strategy profile $s = (s_1, s_2, ..., s_{10})$. The payoff $u_i(s)$ of player $i$ is given by

$$u_i(s_i, s_{-i}) := \begin{cases} L_i(t_w) - \beta_{s_i}(x_{i,t_w}) & \text{if } \beta_{s_i}(x_{i,t_w}) = \max_{j \in N}\{\beta_{j,k} : k \leq T\}, \\ 0 & \text{otherwise.} \end{cases}$$

where $t_w$ denotes the submission time of the winning bid and $s = (s_i, s_{-i})$ as is standard.

## A.2. Model calibration: estimation of private orderflow

The model is implemented with Agent-Based Modeling techniques. For technical reasons, we assume that time evolves at discrete time steps of 10ms increments. The state of the auction is updated after the players take their bidding actions simultaneously.

To inform the settings of our model, we use the mempool data (Flashbots, b) and the on-chain data maintained by Flashbots and Dune for the period February 23 to April 7 2024 (Flashbots & Dune).[2] On average, each block contains 143 public transactions, with each transaction valued at approximately 0.00021 ETH, which collectively accounts for nearly 40% of the total block value. Additionally, builders earn an average profit of 0.0066 ETH from winning an auction. In the simulation, we assume that this profit margin is symmetric across all players.

It is worth noting that the available data sources exhibit a certain degree of bias concerning the private orderflow. The on-chain data *only* reveals the private orderflow included by the winning builder. Since that builder wins the auction, we presume that their access to private orderflow surpasses that of other competing builders within the auction. However, the actual access to private orderflow by a builder remains undisclosed, , irrespective of their success in the auction. This lack of information stems from the data not being recorded on-chain and not being available from any Relay Data API. Consequently, we assume that private orderflow access among players is randomly distributed on a domain which we can estimate from on-chain data.

This approach guarantees that regardless of the auction's winner, the volume of private orderflow included in the winning block aligns with expectations set by on-chain metrics, offering a coherent framework for approximating the distribution of

---

[2]We exclude the data of March 13, due to the error caused by the EIP-4844 Dencun upgrade.

access to private orderflow among players. Specifically, the average private orderflow of the top 10 builders contributes between 11.2% and 14.0% of the total transactions in their respective winning blocks. We subsequently infer that a 14.0% inclusion of private orderflow represents the maximal volume achievable by players, i.e., the total transaction number in the private mempool, corresponding to a private orderflow access probability of 100%. An 11.2% inclusion denotes the minimal threshold, equating to an 80% access probability. Thus, we delineate the distribution of $\pi_i$ to be a uniform distribution spanning the interval $[0.8, 1.0]$.

## B. Game Definition

### B.1. Equal latency and prior orderflow access distribution

We begin by analyzing games where all builders have the same latency and prior distribution on private orderflow access. Specifically, for every auction simulation, each player's probability, $\pi_i$ of accessing private orderflow is drawn from the same prior distribution, namely uniform on $[0.8, 1]$.

In this case, we can reduce the size of the underlying game by replacing it with an *anonymous game*, where a player's payoff is invariant to permutations of other players (Wellman et al., 2024). In other words, a player's payoff depends only on the number of other players playing each strategy. Therefore, we can represent a strategy profile by a vector of the number of players playing each strategy, which allows us to reduce the number of strategy profiles to $\binom{10+|S|-1}{10} = 66$.

To store the payoff information, we want to use the *heuristic payoff table* (HPT) (Walsh et al., 2002), where *payoffs of each strategy* are stored as a function only of the number of players using it. However, while the private orderflow access probabilities are equal in expectation (since they are drawn uniformly from the same prior distribution), every realisation of the random access probability to private orderflow can be different. This implies that players can have different payoffs if they interchange their strategies which, in turn, implies that the payoffs of each strategy are not unique. Thus, HPT cannot be directly applied since this game is, in fact, asymmetric (Tuyls et al., 2020).

To overcome this issue, we let the payoff of each strategy be the *average payoff of the players* using it in each strategy profile. Moreover, we further reduce the impact of randomness introduced by each player's access to private orderflow, by letting each player's payoff be the average profit out of 1,000 auction simulations for each strategy profile. This leverages the fact that the players with the same latency using the same strategy tend to have the same payoff (in expectation) due to their private orderflow access being drawn from the same prior distribution.

Formally, consider the HPT, $\mathcal{H} = (\mathcal{N}, \mathcal{U})$, where $\mathcal{N}$ is a matrix of strategy profile representations of dimension $\binom{10+|S|-1}{10} \times |S|$, and $\mathcal{U}$ is a matrix of payoffs of the same dimension. Entry $\mathcal{N}_{k,j}$ in $\mathcal{N}$ describes the number of players choosing strategy $s_j, j \in \{0, 1, 2\}$ in strategy profile $s^k$, and entry $\mathcal{U}_{k,j}$ in $\mathcal{U}$ describes the average payoff of players choosing strategy $s_j$ in profile $s^k$. $\mathcal{U}_{k,j}$ can be given by

$$\mathcal{U}_{k,j} = \begin{cases} \frac{1}{\mathcal{N}_{k,j}} \sum_{i:s_i=s_j} u_i\left(s^k\right) & \text{if } \mathcal{N}_{k,j} > 0, \\ 0 & \text{otherwise.} \end{cases}$$

### B.2. Different latencies

While the previous case captures an idealised scenario, in practice, builders experience different latencies due to variations in their connectivity to searchers and relays. In this part, we consider games where players have the same distribution of private orderflow access but different latencies. Specifically, we examine scenarios involving 5 players with low latency and 5 players with high latency.

This setup, which resembles current practice, allows us to consider the auction game as a *role-symmetric game* (Wellman et al., 2024), in which players are divided into two *roles* based on their latency: $r_l$ for the 5 low-latency players and $r_h$ for the 5 high-latency players. Within each role, the payoff of each strategy is represented by the average payoff of the players within that role adopting that strategy.

Formally, let the role, $r_i$ of each player $i$ be $\in \{r_l, r_h\}$, indicating whether player $i$ belongs to the low-latency group ($r_l$) or the high-latency group ($r_h$). We extend the HPT $\mathcal{H} = (\mathcal{N}^l \times \mathcal{N}^h, \mathcal{U}^l \times \mathcal{U}^h)$. $\mathcal{N}^l \times \mathcal{N}^h$ is a matrix of strategy profile representations of the dimension of $\binom{5+|S|-1}{5}^2 \times 2|S|$, where $\mathcal{N}^l$ is a counts matrix for $r_l$ and $\mathcal{N}^h$ is a counts matrix for $r_h$. $\mathcal{U}^l \times \mathcal{U}^h$ is a matrix of payoffs of the same dimension. Entry $\mathcal{N}^r_{k,j}$ in $\mathcal{N}^r, r \in \{l, h\}$, describes the number of players choosing strategy $s_j, j \in \{0, 1, 2\}$ within role $r$ in the strategy profile $s^k$, and entry $\mathcal{U}^r_{k,j}$ in $\mathcal{U}^r, r \in \{l, h\}$ describes the

average payoff of players within the role $r$ choosing the strategy $s_j$ in the profile $s^k$. $\mathcal{U}_{k,j}^r$ can be given by

$$\mathcal{U}_{k,j}^r = \begin{cases} \frac{1}{\mathcal{N}_{k,j}^r} \sum_{i:s_i=s_j,r_i=r} u_i\left(s^k\right) & \text{if } \mathcal{N}_{k,j}^r > 0, \\ 0 & \text{otherwise.} \end{cases}$$

### B.3. Different private orderflow access distributions

In fact, the disparity in private orderflow access between builders is significant. We finally consider games where players have the same latency but different private orderflow access probabilities. Similarly, we examine scenarios involving 5 players with high private orderflow access probability and 5 players with low private access probability and consider the games as role-symmetric games.

Additionally, for all the empirical games above, we consider two distinct scenarios and study the players' incentives under these scenarios: 1) the auction interval varying around 12 seconds ($\sigma = 0.1$), where last-minute players face a 50% chance of successfully revealing their bids before the auction closes, and 2) the auction interval being fixed to 12 seconds ($\sigma = 0$), i.e., relay enforcement of rejecting bids after 12 seconds, where last-minute players typically bid at the very end of the auction interval.

## C. Impact of Latency: Profile Comparison

When high-latency players shift to the naive strategy in the optimal profile, switching from the adaptive to the naive strategy may slightly improve the win rate for low-latency players. However, it reduces their profitability, thereby lowering their payoff overall. To quantify this trade-off, we conducted a pairwise comparison between the control profile where all players adopt the naive strategy and the optimal profile. Figure 6 presents the comparison between the average win rates and payoffs of low-latency players in the optimal profile and those in the control profile under varying latency differences. Our simulation results reveal that under varying latency differences, transitioning from the adaptive to the naive strategy leads to an average increase in win rate of 0.38% for low-latency players, but also results in an average reduction in payoff (average profit per auction) of 0.65%. Conversely, for the high-latency naive players, with low-latency players playing the adaptive strategy in the optimal profile, they gain both a higher win rate and a higher payoff. Figure 7 presents the comparison between the average win rates and payoffs of high-latency players in the optimal profile and those in the control profile under varying latency differences. This observation further supports the stability of the optimal profile and the success of the involved strategy choices by the players.

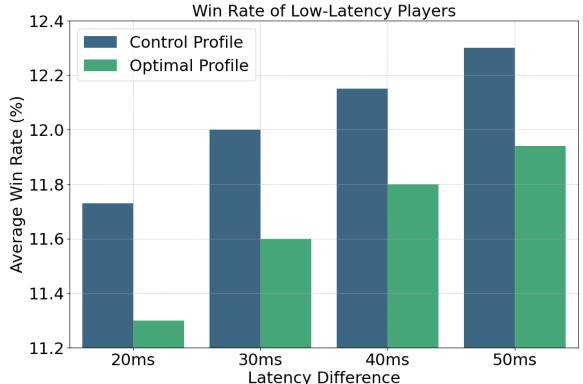
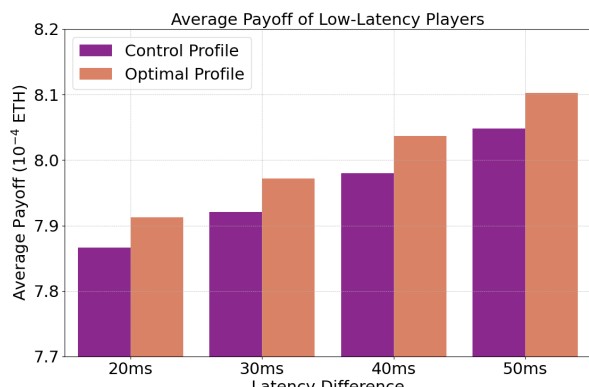

*Figure 6.* The average win rate (left) and average payoff (right) of low-latency players in the optimal profile (adopting the adaptive strategy) in comparison with the control profile (adopting the naive strategy) under varying latency differences starting from 20ms.

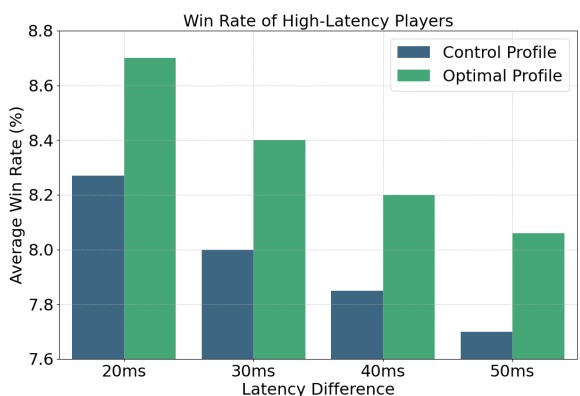
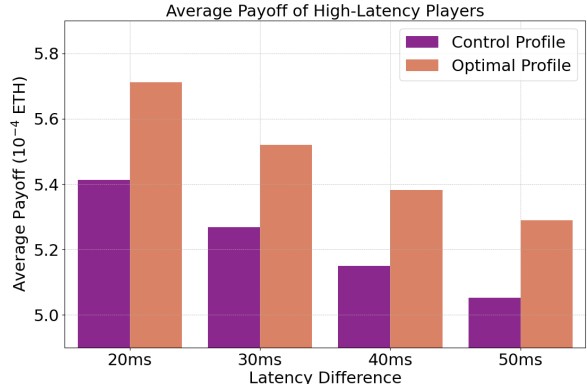

*Figure 7.* The average win rate (left) and average payoff (right) of high-latency players in the optimal profile (adopting the naive strategy) in comparison with the control profile (adopting the naive strategy) under varying latency differences starting from 20ms.

## D. Discussion

### D.1. Limitation

#### D.1.1. MODEL, STRATEGY SPACE, AND GAME DEFINITION

The results and analyses presented in this paper are based on and constrained by the model, calibration methods, and definitions of the strategy space and games. To ensure tractable equilibria in the games with available computational resources, we analyzed the games as *only* one auction with limited information asymmetries (e.g., private orderflow and profit margin) and strategy choices among builders. Due to this constraint and the lack of data, we applied the estimation metrics described in Section A.2 and assumed symmetry among builders for these variables.

The bidding strategies were formulated simply: consistently bidding the full valuation (naive strategy), marginally outbidding when feasible (adaptive strategy), and a last-minute strategy specialized for scenarios under relay enforcement. Whilst these strategies are empirically validated (Wu et al., 2024), it is conceivable to imagine that builders' strategic behaviors and MEV-Boost auction dynamics are far more complex. Our results provide limited insights if more asymmetric information, a larger strategy space, and a series of consecutive auctions are considered. Despite these limitations, our analyses shed light on builders' incentives for strategic bidding in MEV-Boost auctions. We showed that, under idealized conditions of a builder market, builders are incentivized to marginally outbid each other rather than bidding their full valuation, which contributes to a low auction efficiency. We also demonstrated latency improvements and orderflow access advantages can influence builders' strategic bidding incentives.

#### D.1.2. RELAY ENFORCEMENT

In our simulation, we realize the relay enforcement of rejecting new bids after the beginning of the slot by setting the simulated auction interval to exactly 12 seconds, i.e., the duration of a slot. However, despite this enforcement being honestly conducted by the relay to eliminate the proposer's incentive to delay their `getHeader` request, the actual block proposal can still occur sometime after the beginning of the slot. This means the auction interval for the next block will be shorter than 12 seconds. Nevertheless, the crucial factor affecting builders' incentives for choosing strategies is their knowledge of a fixed auction termination time. Even if the actual auction interval might occasionally be shorter, our findings provide valuable directional insights into builders' incentives for strategic behaviors under such conditions.

### D.2. Present and Future Challenges

Currently, while the builder market is highly competitive and the MEV-Boost auction mechanism efficiently captures MEV considering block subsidization (Yang et al., 2024), the market remains centralized and dominated by a few large builders. Based on our analyses, we here discuss the present and future challenges to achieve a decentralized yet competitive builder market.

The primary challenge lies in the vertical integration between different entities across the MEV supply chain. For builders,

vertical integration directly impacts the frequency and volume of orderflow they can access and bid on, thereby enhancing their performance in MEV-Boost auctions and causing centralization. Our results partially explain why vertical integration is attractive to builders by analyzing their incentives for strategic bidding: they are incentivized to outbid marginally to maximize profits through latency improvements, and can dominate the market with greater orderflow access.

For searchers and orderflow providers, the motivation for pursuing vertical integration is fundamentally due to the lack of a *trustless* and *private* mechanism for orderflow distribution between orderflow providers and builders. To safeguard their MEV from being stolen, orderflow providers prefer off-chain exclusive deals with builders based on their reputation in a trusted and permissioned manner. Although OFAs, such as MEV-Share (Flashbots, c) and MEV Blocker (MEV-Blocker), offer orderflow distribution and MEV protection for providers, they are operated by trusted parties. There is also an inherent risk of builders not behaving as expected. A solution under exploration is SUAVE (Flashbots, d;e), a system that operates within Trusted Execution Environments (TEEs) like Software Guard Extensions (SGX). By leveraging TEEs and cryptography, SUAVE intends to enable open but private orderflow distribution for all builders while minimizing the need for trust in these trusted parties and ensuring the confidentiality of private order flow information. However, challenges such as covert channels still exist (Flashbots, a).

Nevertheless, even if a trustless mechanism were available to distribute orderflow equally and privately among all builders, our results indicate that the current MEV-Boost auction mechanism would not efficiently capture MEV under such conditions as builders are incentivized to collude and together increase their bids marginally rather than bidding their full valuation. We believe the builders should be incentivized to bid their full valuation and the auction mechanism should efficiently capture MEV. Otherwise, we question the necessity of having an auction. Furthermore, proposers might be incentivized to fall back to local block production and engage in off-chain deals directly with orderflow providers to increase their MEV revenue, as they cannot secure satisfying bids from colluding builders via the auction. Such bypassing behavior undermines the PBS framework, negating its intended purpose of enhancing validator decentralization. This observation raises the critical open question of whether changes are needed in the block-building auction mechanism and necessitates further research into whether alternative designs could offer improvements.

## E. Related Works

**Builders' strategic behaviors in MEV-Boost auctions.** To the best of our knowledge, the study on builders' strategic bidding behaviors in MEV-Boost auctions starts from (Neuder), where the authors identified certain behaviors such as bid erosion and bid shielding enabled by the relay's bid cancellation feature. Subsequent analyses by (Thiery, a;b) observed and formulated varied bidding behaviors among different builders in MEV-Boost auctions, confirming the use of diverse strategies. Expanding on these foundational studies, (Wu et al., 2024) introduced a game-theoretic model for MEV-Boost auctions alongside four distinct bidding strategies, and conducted simulations on fixed strategy profiles to assess the effects of latency, orderflow, and auction design aspects on the performance of both builders and their strategies. Building upon these models, our work complements these studies by analyzing builders' incentives for strategic bidding across various scenarios within MEV-Boost auctions.

**Latency and timing games.** Latency serves as a crucial infrastructure component of the PBS mechanism and is of significant influence for both builders and proposers. (Pai & Resnick, 2023; Wu et al., 2024) demonstrated that a latency disadvantage adversely affects builders' bidding performance. (Schwarz-Schilling et al., 2023; Natale & Moser, 2024) investigated strategic behaviors by proposers in MEV-Boost auctions, analyzing timing games in which proposers delay the auction termination and increase profitability. (Öz et al., 2023) analyzed how timing games affect consensus stability, and (DataAlways, 2024) further investigated how timing games affect fork rate. Potential mitigation methods for timing games are discussed in (Schwarz-Schilling & Nueder). Our work further shows the impact of latency on builders' incentives for strategic bidding. We also investigate one of the mitigations suggested in (Schwarz-Schilling & Nueder), which involves the relay enforcement of rejecting further bids after the beginning of the slot, and its effects on builders' strategic bidding in the auction.

**MEV Supply chain and private orderflow.** Recent literature has shed light on the status of the MEV supply chain and private orderflow under the current PBS framework. (Wahrstätter et al., 2023) presented the monetary flow between different parties within MEV-Boost auctions, highlighting a positive correlation between the bid value and bid arrival time. (Kilbourn, a;b) discussed the centralizing effect of private orderflow and orderflow auctions on the current block-building market. (Gupta et al., 2023; DataAlways) demonstrated that integrated builders are more likely to win in OFAs when their integrated searchers can provide high-value exclusive orderflow (e.g., CEX-DEX arbitrages), contributing to better performance in

MEV-Boost auctions. Subsequent empirical analysis (DataAlways) showed that integrated builders win in most of the high-value OFAs. (Yang et al., 2024) assessed the competitiveness and efficiency of the MEV-Boost auction, and identified private orderflow as a significant entry barrier to the builder market. Moreover, (Lu) discussed the Ethereum's evolving orderflow landscape, emphasizing the challenge of balancing increased competition with market decentralization. Our results suggest that, under ideal conditions of a builder market that potentially leads to decentralization (builders have similar latency and orderflow access), instead of competing by bidding their full valuation, builders are incentivized to collude by increasing their bids marginally, contributing to the MEV-Boost auction mechanism being inefficient for MEV capture.

