# OpenReview forum: "To Compete or To Collude: Builder Incentives in MEV-Boost Auctions"
_ICML.cc/2024/Workshop/Agentic_Markets — Agentic Markets @ ICML'24 Poster_

### Official Review · Reviewer_UcYC · 2024-06-15
**The paper provides valuable empirical insights into builders' strategic bidding in MEV-Boost auctions, though it requires clearer writing and better explanations of key concepts and methods.**

**Rating:** 6
**Confidence:** 3

**Review:**

Summary:

The paper analyzes builders' strategic bidding in MEV-Boost auctions within Ethereum's PBS framework, finding that builders are incentivized to collude rather than compete when they have uniform latency, leading to low auction efficiency. It also explores the effects of latency advantages, private orderflow access, and relay enforcement policies on builders' incentives and auction outcomes.

The paper makes interesting empirical analyses that can help inform the development of the PBS system. At the same time, the writing is generally a bit lacking in my view, being overly verbose some places and lacking definitions and explanations in others, and generally assuming too much prior knowledge about the Ethereum ecosystem and the specific methods they used to analyze it. I will weakly recommend acceptance and encourage polishing the writing a bit if accepted.

Claims:
- With uniform latency, the builders collude (low efficiency)
- With non-uniform latency, the efficiency is better
- Enforcing a time limit on bids improves efficiency by forcing builders with a latency advantage to bid truthfully

Strengths:
- The three claims to be studied (auction dynamics with uniform latency, non-uniform, and with enforcement) are clearly laid out
- The goals are addressed in the experiments section
- The analyses in 4.2 were very interesting and led to the counterintuitive conclusion that efficiency might increase with latency differences
- Same with analyses in 5., nice tests and a good finding for a suggestion of how to improve auction efficiency

Weaknesses:
- The introduction could use some writing work. Some examples: The language is at times a bit hard to follow, and e.g., the very first sentence spans 7 lines. It's stated that auctions start and end at the start and end of slot $n-1$, but it's unclear what the slot is in relation to a block. It's stated that the proposer calls `getHeader` without making it clear what this means. It's stated without a source that 30 builders are active, but only several builders dominate. It's unclear how the timing games can lead to profits, and I'm confused what slot it is referred to in "...such as rejecting builder bids after the beginning of the slot."
- Section 3: It's not clear to me why we need all this complexity, $3^10 = 59,049$ is not such a big number? I'm sure there's just something I'm missing.
Section 4: It seems the authors are assuming too much knowledge about the specific methods used, e.g., evolutionary stability is used without definition. I think it would be helpful to have had the bidding strategies defined or explained briefly.

Notes and questions:
- Section 2 and 3 are both very short and could just be merged?
- In the setting where all bidders have the same latency, won't the bids over time converge to the signal value as they marginally overbid each other round after round?

---

### Official Review · Reviewer_PZhd · 2024-06-15
**This paper employs empirical game-theoretic analysis to examine builders’ incentives for strategic bidding in MEV-Boost auctions under PBS framework.**

**Rating:** 6
**Confidence:** 4

**Review:**

Summary: This paper studies three different strategies for 10 prominent builders in the current Ethereum Framework. First, it claims that the builders would earn more under collusion (Adaptive strategy) than if they deployed the naive strategy. This collusion phase seems to be representative of an ascending auction, where builders incrementally increase their bids to achieve an equivalence to a second-price auction in a single-round auction that was supposed to be a first-price auction. The paper then introduces more nuances where all builders have different latency and receive private order flow transactions. Even under latencies of the order of 10 ms, using the adaptive strategy remains dominant, even though high latency rate players suffer more than lower latency nodes and would have less than a 5% chance of winning. However, if the latency is greater than 10 ms, the player returns to the naive strategy. The paper then talks about differences in order flows and conclude that builders with higher private order flow are more likely to follow the adaptive strategy.

Finally, the paper talks about Relay Enforcement. In this, the cut-off the auction at the start of the slot instead of until proposer calls getHeader. Under such a scenario, the last minute bidding strategy becomes very viable, since there is no risk of missing the deadline for the auction (since end is determined by a getHeader call from proposer in other case). In this, the high latency players now have an incentive for a last minute bid instead of a naive strategy, thereby disrupting the earlier equilibria. The low latency players thus need to adapt their strategy to remain competitive. This leads to safety under the non-ideal conditions that exist in the current market where some builders would have a higher latency compared to others.

Remarks:
- "players can independently decide their bidding strategy, which results in $3^{10}$ distinct strategy profiles" -> Does it matter which player chooses what profile? Or does it matter only that $n_1$ players use strategy 1, $n_2$ players use strategy 2, and the rest use strategy 3?
- Collusion case: Wouldn't it be better for the collusion to publically announce their value along with a small bid? This way, if someone has a lower value, they can choose to back out, leading to even higher profit for the winning builder. This would also require somewhat of a proof that the builder has the value it is claiming.
- The last minute strategy is under-discussed in the setting where all parties have same latency and private order flows. In Relay Enforcement, it is mentioned that last minute strategy has a chance of failure, but is not well explained.
- Collusion action described in the paper is not really a collusion action, and would be great to change the name. In general, auction theory suggests that under collusion of all parties the winning bid would always be $\epsilon$.

Overall I like the contribution by the authors and note that the problem being studied is fairly important. Relay Enforcement Strategy has potential to be a good competition inducing protocol and ensure that bids are placed as desired from the PBS Auction.